# Fluoride Release, Recharge, and Mass Stability of Restorative Dental Materials: An In Vitro Study

**DOI:** 10.3390/dj13100438

**Published:** 2025-09-23

**Authors:** Md Sofiqul Islam, Vivek Padmanabhan, Ghaid Koniali, Mohannad Zain Alabdin, Smriti Aryal Ac, Nada Tawfig Hashim, Mohamed Ahmed Elsayed, Muhammed Mustahsen Rahman

**Affiliations:** 1Department of Operative Dentistry, RAK College of Dental Sciences, RAK Medical and Health Sciences University, Ras Al Khaimah P.O. Box 11172, United Arab Emirates; 2Department of Pediatric Dentistry, RAK College of Dental Sciences, RAK Medical and Health Sciences University, Ras Al Khaimah P.O. Box 11172, United Arab Emirates; vivek.padmanabhan@rakmhsu.ac.ae; 3Department of Oral and Craniofacial Health Sciences, College of Dental Medicine, University of Sharjah, Sharjah P.O. Box 27272, United Arab Emirates; saryalac@sharjah.ac.ae; 4Department of Periodontology, RAK College of Dental Sciences, RAK Medical and Health Sciences University, Ras Al Khaimah P.O. Box 11172, United Arab Emirates; nada.tawfig@rakmhsu.ac.ae (N.T.H.); mustahsen@rakmhsu.ac.ae (M.M.R.); 5Department of Endodontics, RAK College of Dental Sciences, RAK Medical and Health Sciences University, Ras Al Khaimah P.O. Box 11172, United Arab Emirates; mohamed.elsayed@rakmhsu.ac.ae; 6Department of Endodontics, Faculty of Dentistry, Assiut University, Assiut 71516, Egypt

**Keywords:** glass ionomer cement, resin modified glass ionomer, GIOMER, resin composite, fluoride release, fluoride recharge, water sorption, solubility, mass stability

## Abstract

**Background/Objectives**: Fluoride ion plays a crucial role in protecting teeth against caries by re-mineralizing the caries lesion. The objective of this study was to quantify and compare the fluoride release and recharge of restorative dental materials and their correlation with mass stability. **Methods**: For this study, 5 × 5 × 2 mm blocks were prepared from GIC, RMGI L, GIOMER, Resin Composite, and RMGI R using a customized silicone index. The amount of fluoride released from each material was quantified using a fluoride electrode at 0 h, 1 day, 3 days, 1-week, and 2-week periods. The fluoride recharge of each material was calculated by quantifying the amount of fluoride uptake from high concentration fluoride solution over a period of 1-week. The mass stability of the materials was measured be quantifying the weight loss/weight gain during fluoride release and recharge phase. The correlation of fluoride release/recharge with weight loss/gain were analyzed using Pearson correlation. **Results**: One-way ANOVA showed a statistically significant difference in the amount of fluoride released from each group (*p* < 0.05). The maximum amount of fluoride release was observed on the 3rd day in all the groups except the GIC group, which showed an ascending concentration of fluoride release till 2 weeks. One-way ANOVA showed statistically significant differences in weight loss/gain among the rested group (*p* < 0.05). GIC showed the highest amount of weight loss and weight gain among the tested materials. **Conclusions**: The GIC material has the highest fluoride release and RMGI L has the highest fluoride recharge capability. The conventional GIC showed the least mass stability during fluoride release/recharge.

## 1. Introduction

Restoration of damaged tooth structure with a functionally and esthetically acceptable material is a routine dental procedure. Restoring the damaged tooth structure at its early stage can restore the esthetic and function of the tooth and protect the vital structure of the tooth [1]. Currently available restorative materials can provide durability and patient satisfaction. However, these materials have gone through extensive research and development both in laboratory and clinical settings [2]. The innovation of dental amalgam filling materials in the 19th century made it possible to provide durable permanent restorations at an affordable cost [3]. Since their invention, dental amalgams have been subject to many controversies concerning mercury toxicity and their limitations as an esthetic restorative material [4,5].

To meet the esthetic demand, resin-based composite materials were introduced in the 1960s by Professor Bowen, which could match various tooth colors [6,7]. However, these materials were reported to have poor wear resistance and color stability. The introduction of micro-hybrid and nano-hybrid fillers significantly improved the mechanical properties and wear resistance [8,9,10]. Despite having superior esthetic and durable mechanical properties, resin composites have limited capability to prevent secondary caries [11,12]. This limitation rendered resin composites relatively contraindicated for patients with high caries risk. The quest for restorative material that meets all needs continues.

Glass ionomer-based dental cement (GIC) was developed in 1972 by Wilson and Kent. Soon after its invention, its popularity grew very rapidly due to its ability to release fluoride ion and chemically bind to the tooth structure [13,14]. Fluoride is a naturally occurring mineral that plays an important role in preventing tooth decay. By promoting re-mineralization, it strengthens the enamel and makes it more resistant to bacterial attachment and acidic dissolution of the tooth [15,16]. Utilizing its caries-preventing properties, a wide range of glass ionomer cements were manufactured. To utilize the advantage of fluoride release from glass ionomers, it is often used along with a resin composite using open and close sandwich techniques [17]. Due to their inferior physical properties compared to composite resins, glass ionomers are generally not suitable for use in high occlusal stress-bearing areas [18]. Additionally, limited color availability limits the use of glass ionomers as an esthetic restorative material [19]. To improve mechanical properties, resin-modified glass ionomers (RMGI) were introduced. The superior mechanical properties compared to the conventional glass ionomer and various shade availability restored their status as permanent restorative materials. Alongside resin-modified glass ionomers, a mixture of composite and glass ionomers, namely compomers and GIOMERs, were introduced in the market to meet the demand of caries-preventing restorative materials [20,21].

Past studies have revealed that the fluoride ion release from glass ionomers significantly reduces after robust release for a few days. The amount of fluoride ion release after this robust release is insufficient to prevent caries [22,23]. To achieve an optimal caries prevention effect, it is essential that the fluoride release persists long after placement. Based on that, glass ionomer-based restorative materials can be recharged periodically from external resources. To meet these criteria the filler technology of glass ionomers, compomers and GIOMERs has been redesigned to add fluoride recharge capability [24,25]. Several previous studies have been conducted to measure the fluoride release and recharge of these materials. However, the fluoride release and recharge properties of these materials and their association with physical or structural stability have not been well documented. Correlating fluoride behavior with the mass stability of restorative materials is essential for understanding their long-term clinical performance, especially in preventing secondary caries and maintaining structural integrity [26]. To fill this research gap, it is essential to measure the amount of fluoride release/recharge of restorative dental materials and correlate these properties with mass stability.

In the current in vitro study, we aimed to measure the fluoride release/recharge and mass stability of contemporary restorative materials. The specific objectives were as follows:-To quantify the fluoride release/recharge of GIC, RMGI, GIOMER, and resin composite materials using high/low fluoride concentration storage media;-To quantify the mass stability using a water sorption and solubility test;-To correlate the fluoride release/recharge with mass stability.

## 2. Materials and Methods

**Study design:** This in vitro study protocol was reviewed by the research and ethical committee of Ras Al Khaimah College of Dental Sciences and obtained approval (RAKMHSU-REC-009-2022/23-UG-D, approval date: 27 October 2022) prior to starting the experiment.

**Sample size calculation:** The number of required samples for this study was calculated using statistical software G*Power 3.1.9.7. A pilot experiment was conducted to determine the Cohen’s d effect size. The obtained effect size was validated with a similar previously published article [27]. The required sample size for the ANOVA statistical test was determined using a priori power analysis. The parameters used for the calculation were as follows: effect size f = 0.60, confidence level of 95%, statistical power (1 − β) = 0.90, and numerator degrees of freedom (df) = 4. This analysis indicated that 10 specimens were required for each group (*n* = 10).

**Preparation of materials:** Several 5 mm × 5 mm × 2 mm thick material blocks were prepared using a silicone rubber mold. In group 1, the conventional glass ionomer cement shade A2 was used. The powder and liquid were mixed as per the manufacturer’s instruction. After obtaining a homogeneous texture, the mix was packed into the silicone mold and covered with a glass slide to achieve uniform thickness and a smooth surface. In group 2, RMGI L was injected into the silicon mold and covered with a glass slide. Thereafter, a curing light was applied using a Paradigm™ Deep Cure LED Curing Light (3M Oral Care, St. Paul, MN, USA) for 40 s to polymerize it. In group 3, GIOMER restorative materials were packed and polymerized using the same technique. In group 4, nano-hybrid composite blocks were prepared following the same protocol. In group 5, RMGI R was mixed in an amalgamator, injected into the mold, and polymerized as per the manufacturer’s instructions. A total of 50 specimens were prepared following these techniques. The details of the experimented materials shown in Table 1.

**Fluoride release and mass stability:** The material blocks were then placed in individual containers with 40 mL of distilled water and incubated at 37 °C. The fluoride concentration in the solution was measured at 0 h, 1 day, 3 days, 1 week, and 2 weeks using a handheld fluoride electrode (Extech FL700, Teledyne Technologies Incorporated, Thousand Oaks, CA, USA). The electrode was immersed in TISAB buffer to stabilize the ionic strength and pH of a sample and then calibrated using standard solution prior to each measurement. The incubating solution was refreshed after each measurement. The total fluoride release was measured in accumulation solution. Along with fluoride, the weight of each material block was measured by a precise analytical balance (U.S. Solid 220 × 0.0001 g Analytical Balance, 0.1 mg Lab Balance Digital Precision Scale, U.S. Solid, Cleveland, OH, USA) at a constant temperature of 23 °C and 55% relative humidity before the start of the experiment (W1) and at the same time intervals. After 2 weeks of incubation, the specimens were dehydrated in a desiccator for 24 h in the presence of silica gel to remove moisture from the material block. The dry mass weight W2 was measured and the weight loss (WL) during fluoride release was calculated using the following formula: WL = (W1 − W2)/W1.

**Fluoride recharge and mass stability:** After 2 weeks of fluoride release, the material blocks were then incubated in individual containers with 40 mL of 10 ppm solution and incubated at 37 °C for 1 week. The fluoride solution was prepared by mixing NaF powder in distilled water. The fluoride concentration of the prepared solution was confirmed using electrode prior to immersion of the materials blocks. The weight of the material blocks (W3) and fluoride concentration in the incubating solutions were measured after 1 week of incubation. The fluoride recharge of the materials was calculated by measuring the fluoride concentration reduction in the incubating solution from the baseline. The weight change (WG) in fluoride recharge phase was calculated using the following formula: WG = (W3 − W2)/W2.

**Statistical analysis:** The data of fluoride release, recharge, and weight loss/gain were analyzed using statistical software (SPSS.24.0, IBM, New York, NY, USA). To evaluate the data distribution, descriptive statistics analyses were performed. The normality of the data was evaluated using the Shapiro–Wilk test. Parametric analysis was employed based on data distribution results. A two-way ANOVA was conducted to evaluate the effect of materials and incubation time on the amount of fluoride release/recharge and weight loss/gain. Multiple comparisons among the tested groups and incubation time were performed at 95% confidence level. The correlation between fluoride release/recharge and weight loss/gain were performed using the Pearson correlation test. The graphs were generated in GraphPad Prism 10.2.3.

## 3. Results

The two-way ANOVA showed that both the materials and time have a statistically significant effect on the amount fluoride release (*p* < 0.05). A statistically significant interaction (*p* = 0.001) between the factors (materials* time) was observed. The maximum amount of fluoride release was observed on the third day in all the groups except the GIC group, which showed an ascending concentration of fluoride release until 2 weeks. The amount of fluoride release of each material at different time intervals is shown in Figure 1.

Among the tested materials, GIC showed the maximum amount of fluoride release, followed by GIOMER, RMGI R, and RMGI L. The least amount of fluoride release was observed in composite group. The collective amount of fluoride release by each material shown in Figure 2.

All the tested materials showed a certain amount of weight loss upon dehydration after 2 weeks of fluoride release. The mean weights of each material at different time intervals during fluoride release are shown in Figure 3.

The average fluoride release from each material and the average weight of each material block at different time intervals are summarized in Table 2.

Among the tested materials, GIC showed the maximum amount of weight loss, which was statistically significant compared to other groups (*p* = 001). Groups identified with the same alphabet letter are statistically insignificant. However, the remaining groups were statistically insignificant among them (*p* = 0.538). The weight loss of each material is shown in Figure 4.

The amount of fluoride recharge significantly varied among the tested materials (*p* = 0.001). The RMGI L group showed the maximum amount of fluoride recharge followed by GIOMER and composite. The GIC and RMGI R groups showed the lowest amount of fluoride recharge. The amount of fluoride recharge by the tested materials is shown in Figure 5.

For weight gain during the fluoride recharge phase, there was no significant difference among the tested groups, except for the GIC group which showed a significantly higher weight gain compared to other tested groups. The amount of weight gain during fluoride recharge phase is shown in Figure 6.

A Pearson correlation assay showed that overall fluoride release has a strong correlation with the weight loss of the restorative materials (r = 0.724, *p* = 0.0001). However, fluoride recharge showed a moderate correlation with weight gain (r = −0.434, *p* = 0.030). Among the tested materials, the fluoride release of the composite showed a strong negative correlation (r = −0.827) with weight loss. In the case of fluoride recharge, the composite showed a strong negative correlation (r = −0.807) and RMGI R showed a strong positive correlation (r = 0.849) with weight gain. The correlation of fluoride release/recharge with weight loss/gain of each tested material is shown in Table 3.

## 4. Discussion

Together with their relationship to mass stability, the results of this study offer important new perspectives on the release of fluoride and recharge behavior of several restorative dental materials. The results show that the fluoride release and recharge characteristics as well as the related variations in material weight were strongly influenced by both the type of material and incubation period.

Apart from typical glass ionomer cement (GIC), which showed a steady rise in fluoride release over two weeks, the study found that the greatest fluoride release for most investigated materials occurred on the third day. This is consistent with earlier studies showing an early surge in fluoride release from glass ionomer-based materials followed by a slow decrease over time [26,28]. GIC’s high fluoride release fits its natural ionic matrix, which permits constant ion exchange with the surrounding medium [29]. Moreover, the analysis by Garcia, I. M. et al., 2016, revealed comparable trends of fluoride leakage, therefore strengthening the validity of our results [30].

Resin-modified glass ionomer restoratives (RMGI R and RMGI L) showed reduced fluoride release as compared to GIC, therefore confirming the findings of earlier research showing that resin component polymerization in RMGI limits fluoride diffusion [31]. Pre-reacted glass ionomer fillers were included in GIOMER, which showed fluoride release in between GIC and resin-modified materials. This result aligns with that of Itota T. et al. [24] who found that GIOMER materials release fluoride in a more regulated way than traditional GIC. Composites had negligible intrinsic fluoride-releasing characteristics unless modified with fluoride-containing fillers and composite resin materials showed the lowest fluoride release [24]. Although composite resins have better mechanical and esthetic qualities, their lack of ability to release appreciable fluoride reduces their anticariogenic action [32].

The bulk stability of these materials during fluoride leakage was another focus of the work. Observed dehydration-induced weight loss among all tested materials revealed material breakdown over time. The GIC group showed the most weight loss, which fits its increased water sorption qualities and fluoride release [33]. A strong positive association between fluoride release and weight loss indicated by the correlation analysis shows that the greater the fluoride release, the more the material matrix degrades. This finding conforms to those of earlier studies showing that ion leaching may cause structural changes in fluoride-releasing compounds [34].

On the other hand, the lower weight loss shown in RMGI indicated that the polymerization of resin components gave enhanced resistance to degradation while preserving some fluoride release. These results are in accordance with earlier research which shows that compared to traditional GIC, the resin component in RMGIs adds to increased durability and less solubility [35].

After two weeks of fluoride release, the materials were exposed to a fluoride-containing environment to assess their recharging capacity. The results showed that among the examined materials, the capacity to absorb and then release fluoride differed greatly. Resin-modified glass ionomer in a liner form (RMGI L) had the best fluoride recharge capacity; GIOMER and composite resin followed. The phenomenon of high fluoride recharge by RMGI aligns with the previous study by Rai, S. et al., 2019, where a similar finding was observed [36]. Conventional GIC and RMGI R, however, demonstrated reduced fluoride recharge capacity; this might be explained by their early depletion of fluoride supplies during the early burst release phase. This outcome is contrary to the previous study by Naoum, S. et al., 2011, that showed that conventional GIC has a high fluoride recharge capability [37]. The amount of fluoride release and recharge can be influenced by several factors like release media, pH cycle, brushing abrasion, and surface properties of the materials, etc. Variations in experimental settings, fluoride concentrations applied during recharge, or changes in material formulas between manufacturers could help to explain this variance [38].

All studied materials showed weight increase during the fluoride recharge phase; however, only in the GIC group was this statistically significant observation noted. The correlation study found a modest negative link between fluoride recharge and weight increase, meaning that materials with increased fluoride absorption did not always experience corresponding mass gain. There are several previous studies that quantify the fluoride recharge of different restorative materials [39,40,41]. However, there are limited/insufficient studies that evaluate the fluoride recharge and its correlation with the weight gain/loss of the restorative materials. Further research is required to conduct a comparison with the findings of the current study. The degradation of resin-based materials is multifactorial. The dissolution of the resin matrix plays an important role in this phenomenon. Although our study finds a correlation with fluoride release/recharge, identification and quantification of each ion that is exchanged during this incubation period should be measured using HPLC and SEM-EDX. The bioactivity of these materials significantly affects the re-mineralization capability and strength of the restoration [42].

The oral environment is a dynamic condition where the composition of saliva may vary among individuals [43,44]. This variation may lead to different clinical outcomes of restorative material tested and reported in our study. Another limitation of this study is the lack of established protocol to evaluate fluoride release/recharge of restorative materials in vitro condition. Further research is needed to establish a standard protocol for such experiments. Future research should investigate, considering patient-based factors such as salivary composition, dietary patterns, and oral hygiene routines, the therapeutic consequences of these discoveries in real-world environments.

The results of this study have significant clinical relevance. Particularly in high-load-bearing locations, dental professionals should carefully assess the lifetime of fluoride-releasing materials given their strong link with weight loss. Although GIC showed better fluoride release, its faster rate of breakdown could hinder restoration longevity. With particular regard to light-cured versions, RMGIs seem to offer a good compromise between structural integrity, fluoride release, and recharging. GIOMER materials’ modest fluoride recharge capability makes them an appealing option for long-term caries prevention without appreciable material degradation. This work offers important new perspectives on the mass stability, fluoride release, and recharging of several restorative dental materials. While RMGI and GIOMER materials showed a superior balance between fluoride release and material stability, conventional GIC displayed the highest fluoride release but also the greatest mass loss. While traditional GIC showed low recharge potential, fluoride recharge was most effective in light-cured resin-modified glass ionomers. These results highlight the significance of choosing restorative materials depending on patient-specific demands considering both long-term durability and fluoride release potential. The clinician should also consider the type of restoration, such as whether it is in pediatric, adult, or geriatric patients and whether it is in anterior or posterior sites, while choosing such materials.

## 5. Conclusions

Within the limitations of the in vitro study model, the results of this study concluded that conventional GIC has the highest fluoride release and RMGI has the highest fluoride recharge capability. Conventional GIC shows the greatest degree of weight loss/gain during fluoride release and recharge. Fluoride release has a strong correlation with weight loss and fluoride recharge has a moderate correlation with the weight loss of restorative dental materials. The clinician should carefully choose the type of fluoride-releasing material based on the expected esthetic, functional, and biological requirements of the restoration.

## Figures and Tables

**Figure 1 dentistry-13-00438-f001:**
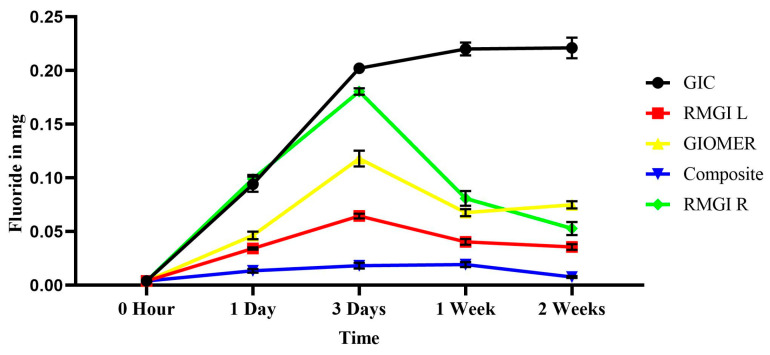
The amount of fluoride release in mg from each material at each time interval.

**Figure 2 dentistry-13-00438-f002:**
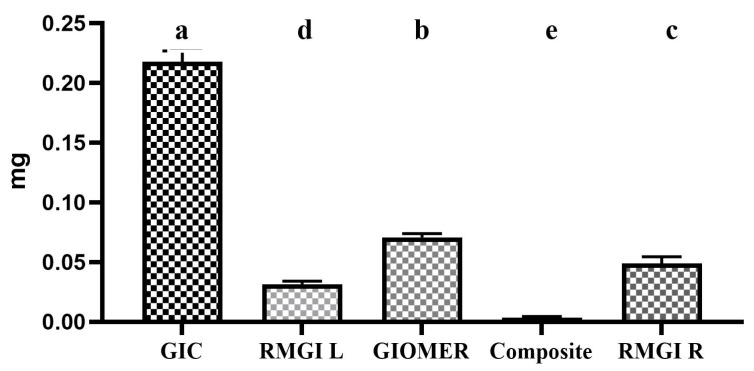
The collective amount of fluoride release by each material. Groups identified with the same alphabet letter are statistically insignificant (*p* > 0.05).

**Figure 3 dentistry-13-00438-f003:**
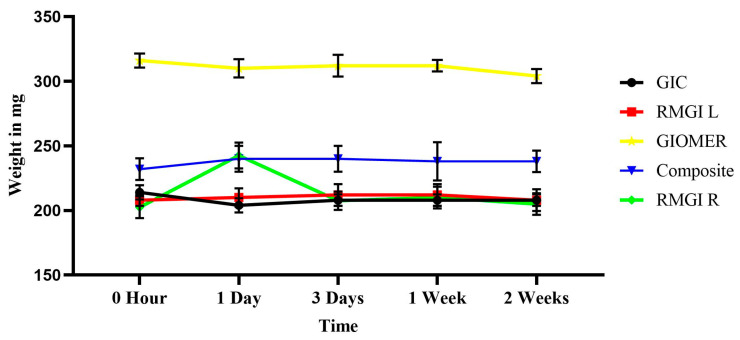
The mean weights of each material at different time intervals during fluoride release.

**Figure 4 dentistry-13-00438-f004:**
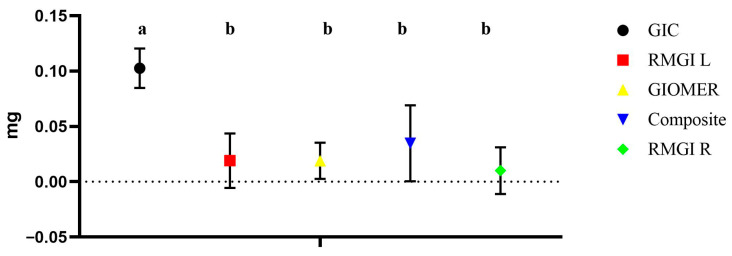
The weight loss of each material during fluoride release phase. Groups identified with the same alphabet letter are statistically insignificant (*p* > 0.05).

**Figure 5 dentistry-13-00438-f005:**
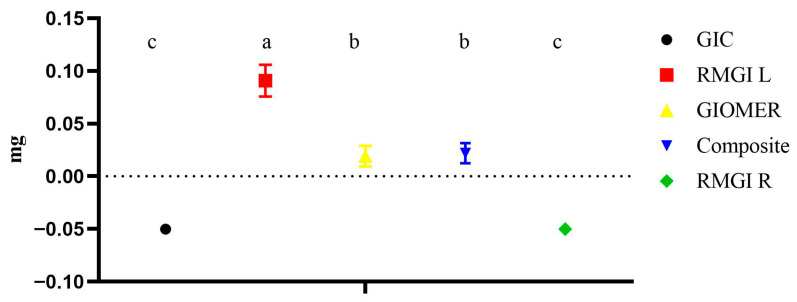
The amount of fluoride recharges by the tested materials. Groups identified with the same alphabet letter are statistically insignificant (*p* > 0.05).

**Figure 6 dentistry-13-00438-f006:**
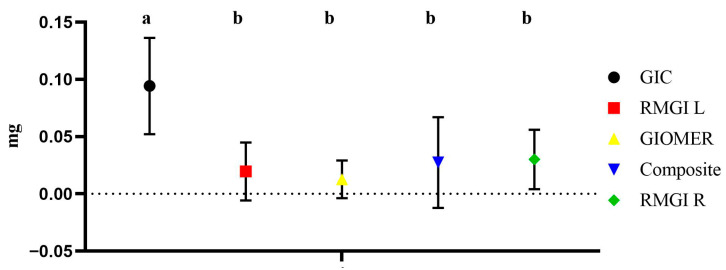
The amount of weight gain by each material during fluoride recharge phase. Groups identified with the same alphabet letter are statistically insignificant (*p* > 0.05).

**Table 1 dentistry-13-00438-t001:** Group distribution and product detail of the tested materials.

Groups	Name of the Materials	Product Details	Lot Number
GIC	Glass ionomer cement	Medifil, Promedica Dental Material GmbH Domagkstrasse 31, 24537 Neumuenster, Germany	2118258
RMGI L	Resin modified glass ionomer liner	Nova Glass GL, IMICRYL, Anatolia, Turkey	221808
GIOMER	Surface pre-reacted glass ionomer (S-PRG) filler-based resin composite	BEAUTIFIL II, SHOFU INC., Kamitakamatsucho, Fukuina, Higashiyama-ku, Kyoto City, Japan	092280
Composite	nano-hybrid resin composite	Z350 XT Universal Restorative, 3M Filtek, 3M Oral Care, St. Paul, MN, USA	10993033
RMGI R	Resin modified glass ionomer restorative	Nova Glass II LC, IMICRYL, Anatolia, Turkey	21E836

**Table 2 dentistry-13-00438-t002:** Fluoride release from each material and the average weight of each material block at different time intervals. Groups identified with same alphabet letter are statistically insignificant.

Fluoride Release from the Materials in ppm					
Time	Materials				
	GIC	RMGI L	GIOMER	Composite	RMGI R
0 h	0.0040 ± 0.000 ^d^	0.0040 ± 0.000 ^d^	0.0040 ± 0.000 ^d^	0.0040 ± 0.000 ^d^	0.0040 ± 0.000 ^e^
1 day	0.0940 ± 0.007 ^c^	0.0342 ± 0.001 ^c^	0.0464 ± 0.003 ^c^	0.0134 ± 0.001 ^b^	0.0990 ± 0.003 ^b^
3 days	0.2020 ± 0.002 ^b^	0.0644 ± 0.002 ^a^	0.1180 ± 0.007 ^a^	0.0182 ± 0.002 ^a^	0.1804 ± 0.002 ^a^
1 week	0.2208 ± 0.006 ^a^	0.0404 ± 0.002 ^b^	0.1180 ± 0.007 ^a^	0.0192 ± 0.002 ^a^	0.0808 ± 0.006 ^c^
2 weeks	0.2216 ± 0.009 ^a^	0.0356 ± 0.002 ^c^	0.0748 ± 0.003 ^b^	0.0076 ± 0.002 ^c^	0.0528 ± 0.006 ^d^
*p* Value	0.001	0.001	0.001	0.001	0.001
Weight of the materials in mg					
Time	Materials				
	GIC	RMGI L	GIOMER	Composite	RMGI R
0 h	214 ± 5.4 ^a^	208 ± 4.4 ^a^	316 ± 5.4 ^a^	232 ± 8.3 ^a^	202 ± 8.3 ^b^
1 day	204 ± 5.4 ^b^	210 ± 7.0 ^a^	310 ± 7.0 ^ab^	240 ± 10.0 ^a^	240 ± 10.0 ^a^
3 days	208 ± 4.4 ^ab^	212 ± 8.3 ^a^	312 ± 8.3 ^ab^	240 ± 10.0 ^a^	210 ± 7.0 ^b^
1 week	208 ± 4.4 ^ab^	212 ± 8.3 ^a^	312 ± 4.4 ^ab^	238 ± 14.8 ^a^	212 ± 8.3 ^b^
2 weeks	208 ± 4.4 ^ab^	208 ± 8.3 ^a^	304 ± 5.4 ^b^	238 ± 8.3 ^a^	208 ± 8.3 ^b^
*p* Value	0.062	0.836	0.084	0.749	0.001

**Table 3 dentistry-13-00438-t003:** The mean fluoride release, weight loss, fluoride recharge, and weight gain of each material. Pearson correlation (r) of fluoride release/recharge with weight loss/gain of each tested material. Groups identified with same alphabet letter are statistically insignificant.

Materials	Fluoride Release	Weight Loss	Fluoride Recharge	Weight Gain
GIC	0.217 ± 0.009 a	0.102 ± 0.017 a	−0.050 ± 0.030 c	0.094 ± 0.042 a
RMGICL	0.031 ± 0.002 d	0.019 ± 0.024 b	0.090 ± 0.015 a	0.019 ± 0.025 b
GIOMER	0.070 ± 0.003 b	0.018 ± 0.016 b	0.019 ± 0.009 b	0.012 ± 0.016 b
Composite	0.003 ± 0.0008 e	0.034 ± 0.034 b	0.022 ± 0.009 b	0.027 ± 0.039 b
RMGICR	0.048 ± 0.005 c	0.010 ± 0.021 b	−0.050 ± 0.0001 c	0.030 ± 0.025 b
*p* value	0.001	0.001	0.001	0.004
Correlation				
Materials	Fluoride release correlation with Weight loss		Fluoride recharge correlation with Weight gain	
GIC	−0.507		0.205	
RMGICL	0.230		0.490	
GIOMER	0.238		0.275	
Composite	−0.827		−0.801	
RMGICR	−0.073		0.849	

## Data Availability

The raw data supporting the conclusions of this article will be made available by the corresponding author upon request.

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
