# Peer review of "Fluoride Release, Recharge, and Mass Stability of Restorative Dental Materials: An In Vitro Study"

_dentistry, 2025, doi:10.3390/dj13100438_

Round 1
Reviewer 1 Report
Comments and Suggestions for Authors
The study addresses a relevant and clinically significant topic, especially for preventive dentistry and restorative material selection in high-caries-risk patients. The inclusion of both fluoride dynamics and mass stability offers a multifaceted view rarely explored in similar studies.
However there are some shortcomings that needs to be addressed.
Language and grammar:
- The manuscript would benefit from grammatical editing. There are minor grammatical inconsistencies and awkward phrasing throughout the text (e.g., "fluoride recharge phage" should be "fluoride recharge phase").
- The manuscript inconsistently uses abbreviations for restorative materials. For example, "RMGIC" is sometimes written in full and sometimes abbreviated without a clear first definition. Introduce all abbreviations consistently in the introduction or methods section and use them uniformly throughout the text.
- Please correct all the typographical/grammatical errors such as: Line 88: “carries prevention” should be should be corrected to “caries prevention”. Line 91: “to Meet” should be “meet” should not be capitalized. Line 92: There is an unnecessary space before “to”.
Introduction:
- The introduction provides a clear chronological overview of restorative materials, from amalgam to resin composites to fluoride-releasing systems. The historical framing is too long. Some parts (e.g., the discussion of dental amalgam and its controversies) could be shortened or streamlined, as they are not directly relevant to the study’s objectives.
- The middle part of the introduction appropriately shifts toward fluoride-releasing materials like GIC, RMGI, GIOMER, and compomers. The clinical importance of fluoride in caries prevention is well stated. However, the transition from general background to specific knowledge gaps is somewhat abrupt and could be more clearly signposted.
- Some references are very old (e.g., Wilson & Kent, 1972) and used without support from more recent clinical evidence.
- There is limited mention of previous studies that analyzed both fluoride dynamics and physical stability together, which is the stated novelty of the current work. Highlighting this research gap earlier would improve the justification for the study.
- The final paragraph introduces the study's aim: to investigate fluoride release/recharge and its correlation with mass stability. This is appropriate, but the rationale could be sharpened. For example, why this correlation matters in a clinical setting? Why comparing multiple material types under standardized conditions is important?
A language revision would improve clarity and readability.
Methodology:
The sample size calculation lacks key details for reproducibility. While the authors state that they used G*Power with f = 0.75, α = 0.05, and power = 0.90, they do not specify the test type (e.g., One-Way vs. Two-Way ANOVA) or the numerator degrees of freedom (df).
This is critical since the analysis uses a Two-Way ANOVA, which requires more complex input and usually a larger sample. The authors should clarify this.
Additionally, the assumed effect size (f = 0.75) is unusually large. Without justification (e.g., previous studies), this risks underestimating the required sample.
The fluoride recharge was performed only once (10 ppm for 1 week). This does not fully reflect the dynamic, repeated exposure seen in clinical conditions (e.g., daily fluoride toothpaste or mouthrinse use). Future studies should simulate repeated short-term fluoride exposure to mimic real-world oral environments.
Surface roughness, porosity, or coating agents may significantly affect both fluoride release and degradation, but these were not accounted for. Future work should consider SEM/EDX or profilometry analysis for a more detailed material characterization.
The authors report using both the Shapiro–Wilk and Kolmogorov–Smirnov tests to assess data normality. This is redundant, as both serve the same purpose. Given the small sample size (n = 10 per group), the Shapiro–Wilk test is more appropriate and generally recommended. The Kolmogorov–Smirnov test is less sensitive for small samples and is not necessary in this context.
Appropriate statistical tools (Two-way ANOVA, Pearson correlation) were applied, and significance levels are clearly reported.
Results.
- Numerical data are primarily presented in graphical format. Inclusion of tabulated raw or mean values with standard deviations would enhance clarity and reproducibility.
- Please use standardized abbreviations consistently across text, tables, and figures to avoid confusion.
- The statement that RMGIL had the highest recharge is interesting, but it's unclear whether this was statistically different from GIOMER and Composite. Clearly state which group differences were statistically significant (e.g., include p-values or letters above bars to indicate post-hoc results).
- Weight loss during fluoride release and gain during recharge are briefly described but not fully interpreted. Comment more directly on the clinical implications (e.g., Does higher weight loss suggest greater material degradation?).
- The manuscript reports both positive and negative correlations between fluoride release or recharge and weight loss or gain. For example, in the composite group, a strong negative correlation is observed between fluoride release and weight loss (R² = –0.827). This suggests that samples releasing more fluoride actually lost less weight, which seems counterintuitive. Typically, fluoride release is expected to be associated with matrix degradation and water loss, leading to greater weight loss. Therefore, a positive correlation would be more biologically plausible. A negative correlation, especially in composites that release very little fluoride, might indicate a different release mechanism, minimal structural impact, or simply statistical noise caused by low variation in the data. The authors should clarify the physical or material explanation behind these unexpected correlation directions, particularly for materials like composites.
Discussion
- The discussion is generally well-structured and follows the logical flow of the results. The authors correctly highlight the most significant findings, such as the superior fluoride release of GIC and the recharge ability of RMGIC, and attempt to interpret them in light of material composition and matrix structure. However, some points remain superficially addressed, and key interpretations lack depth or direct clinical linkage. For instance, the implications of weight loss for material longevity are mentioned, but not fully explored in a clinical context (e.g., marginal integrity, secondary caries risk).
- Some citations are outdated or generic
- Although clinical relevance is briefly addressed, it would be useful to explicitly discuss how the findings might influence material choice in specific clinical scenarios (e.g., pediatric vs. geriatric patients, posterior restorations vs. non-load-bearing areas).
- Limitations: The authors appropriately acknowledge the in vitro nature of the study and mention the need for standardized protocols and patient-based research. This is good, but somewhat generic.
- It would be helpful to also mention specific variables not addressed, such as pH cycling, salivary enzymes, or brushing abrasion, which could significantly impact fluoride dynamics and material stability.
Figures 1–6:
Make sure axis labels and legends are fully self-explanatory; some are slightly unclear.
Author Response
The author appreciates the valuable comments of the reviewer. The comments were helpful in improving the quality of the manuscript.
Comment: The study addresses a relevant and clinically significant topic, especially for preventive dentistry and restorative material selection in high-caries-risk patients. The inclusion of both fluoride dynamics and mass stability offers a multifaceted view rarely explored in similar studies.
Reply: Thank you for the positive feedback and insight.
Comment: The manuscript would benefit from grammatical editing. There are minor grammatical inconsistencies and awkward phrasing throughout the text (e.g., "fluoride recharge phage" should be "fluoride recharge phase").
Reply: Thank you for pointing out the typos. The typos and grammatical errors have been revised accordingly.
Comment: The manuscript inconsistently uses abbreviations for restorative materials. For example, "RMGIC" is sometimes written in full and sometimes abbreviated without a clear first definition. Introduce all abbreviations consistently in the introduction or methods section and use them uniformly throughout the text.
Reply: Thank you for pointing out the abbreviation error. The errors have been revised accordingly.
Comment: Please correct all the typographical/grammatical errors such as: Line 88: “carries prevention” should be should be corrected to “caries prevention”. Line 91: “to Meet” should be “meet” should not be capitalized. Line 92: There is an unnecessary space before “to”.
Reply: Thank you for pointing out the typos. The typos have been revised accordingly.
Comment: The introduction provides a clear chronological overview of restorative materials, from amalgam to resin composites to fluoride-releasing systems. The historical framing is too long. Some parts (e.g., the discussion of dental amalgam and its controversies) could be shortened or streamlined, as they are not directly relevant to the study’s objectives.
Reply: Thank you for the comment. This section has been revised accordingly
Comment: The middle part of the introduction appropriately shifts toward fluoride-releasing materials like GIC, RMGI, GIOMER, and compomers. The clinical importance of fluoride in caries prevention is well stated. However, the transition from general background to specific knowledge gaps is somewhat abrupt and could be more clearly signposted.
Reply: Thank you for the comment. This section has been revised accordingly
Comment: Some references are very old (e.g., Wilson & Kent, 1972) and used without support from more recent clinical evidence.
Reply: Thank you for the comment. The citation has been replaced with a recent one.
Comment: There is limited mention of previous studies that analyzed both fluoride dynamics and physical stability together, which is the stated novelty of the current work. Highlighting this research gap earlier would improve the justification for the study.
Reply: Thank you for the comment. This section has been revised accordingly
Comment: The final paragraph introduces the study's aim: to investigate fluoride release/recharge and its correlation with mass stability. This is appropriate, but the rationale could be sharpened. For example, why this correlation matters in a clinical setting? Why comparing multiple material types under standardized conditions is important?
Reply: Thank you for pointing out an important aspect. This section has been revised accordingly
Comment: A language revision would improve clarity and readability.
Reply: Thank you for the comment. This language has been carefully revised.
Comment: The sample size calculation lacks key details for reproducibility. While the authors state that they used G*Power with f = 0.75, α = 0.05, and power = 0.90, they do not specify the test type (e.g., One-Way vs. Two-Way ANOVA) or the numerator degrees of freedom (df).
This is critical since the analysis uses a Two-Way ANOVA, which requires more complex input and usually a larger sample. The authors should clarify this. Additionally, the assumed effect size (f = 0.75) is unusually large. Without justification (e.g., previous studies), this risks underestimating the required sample.
Reply: Thank you for the comment. This sample size calculation section has been revised, and a rational explanation of effect size has been added.
Comment: The fluoride recharge was performed only once (10 ppm for 1 week). This does not fully reflect the dynamic, repeated exposure seen in clinical conditions (e.g., daily fluoride toothpaste or mouthrinse use). Future studies should simulate repeated short-term fluoride exposure to mimic real-world oral environments.
Reply: Thank you for pointing out an important aspect of the study. We completely agree with the reviewer. To mimic the oral environment, it is essential to employ cyclic fluoride release and recharge rather than a single dosage approach; however, due to the limitation of the electrode method to quantify the fluoride concentration, which measures of single solution at a single point of time, rather than a dynamic plot of multiple specimens in parallel. Another limitation was the comparative nature of this study, where the experimental materials were significantly different from each other in terms of fluoride release and uptake. To ensure the recharge capability of all experimental material, we have chosen a single dosage over a cyclic dosage.
Comment: Surface roughness, porosity, or coating agents may significantly affect both fluoride release and degradation, but these were not accounted for. Future work should consider SEM/EDX or profilometry analysis for a more detailed material characterization.
Reply: Thank you for pointing out another important aspect of these materials. Definitely, that would be an interesting future study.
Comment: The authors report using both the Shapiro–Wilk and Kolmogorov–Smirnov tests to assess data normality. This is redundant, as both serve the same purpose. Given the small sample size (n = 10 per group), the Shapiro–Wilk test is more appropriate and generally recommended. The Kolmogorov–Smirnov test is less sensitive for small samples and is not necessary in this context.
Reply: Thank you for pointing out the error. This section has been revised accordingly
Comment: Numerical data are primarily presented in graphical format. Inclusion of tabulated raw or mean values with standard deviations would enhance clarity and reproducibility.
Reply: Thank you for the suggestion. A table containing numerical data has been added to the manuscript.
Comment: Please use standardized abbreviations consistently across text, tables, and figures to avoid confusion.
Reply: Thank you for the comment. The abbreviations have been revised carefully.
Comment: The statement that RMGIL had the highest recharge is interesting, but it's unclear whether this was statistically different from GIOMER and Composite. Clearly state which group differences were statistically significant (e.g., include p-values or letters above bars to indicate post-hoc results).
Reply: Thank you for the comment. The statistical significance has been revised carefully.
Comment: Weight loss during fluoride release and gain during recharge are briefly described but not fully interpreted. Comment more directly on the clinical implications (e.g., Does higher weight loss suggest greater material degradation?).
Reply: Thank you for the comment. This section has been revised as per the comment.
Comment: The manuscript reports both positive and negative correlations between fluoride release or recharge and weight loss or gain. For example, in the composite group, a strong negative correlation is observed between fluoride release and weight loss (R² = –0.827). This suggests that samples releasing more fluoride actually lost less weight, which seems counterintuitive. Typically, fluoride release is expected to be associated with matrix degradation and water loss, leading to greater weight loss. Therefore, a positive correlation would be more biologically plausible. A negative correlation, especially in composites that release very little fluoride, might indicate a different release mechanism, minimal structural impact, or simply statistical noise caused by low variation in the data. The authors should clarify the physical or material explanation behind these unexpected correlation directions, particularly for materials like composites.
Reply: Thank you for the comment. This section has been revised as per the comment.
Comment: The discussion is generally well-structured and follows the logical flow of the results. The authors correctly highlight the most significant findings, such as the superior fluoride release of GIC and the recharge ability of RMGIC, and attempt to interpret them in light of material composition and matrix structure. However, some points remain superficially addressed, and key interpretations lack depth or direct clinical linkage. For instance, the implications of weight loss for material longevity are mentioned, but not fully explored in a clinical context (e.g., marginal integrity, secondary caries risk).
Reply: Thank you for the comment. This section has been elaborated as per the comment.
Comment: Some citations are outdated or generic
Reply: The citations have been updated.
Comment: Although clinical relevance is briefly addressed, it would be useful to explicitly discuss how the findings might influence material choice in specific clinical scenarios (e.g., pediatric vs. geriatric patients, posterior restorations vs. non-load-bearing areas).
Reply: Thank you for the suggestion. We have added a statement on it in the revised manuscript.
Comment: The authors appropriately acknowledge the in vitro nature of the study and mention the need for standardized protocols and patient-based research. This is good, but somewhat generic. It would be helpful to also mention specific variables not addressed, such as pH cycling, salivary enzymes, or brushing abrasion, which could significantly impact fluoride dynamics and material stability.
Reply: Thank you for the suggestion. We have revised this section accordingly.
Comment: Figures 1–6: Make sure axis labels and legends are fully self-explanatory; some are slightly unclear.
Reply: Thank you for the suggestion. We have revised this section accordingly.
Reviewer 2 Report
Comments and Suggestions for Authors
- Many grammatical and spelling errors, correctable by the authors or the journal’s editorial assistant.
- A table of materials used should be substituted for the descriptions in the text. Include batch numbers. The restorative material names are confusing: RMGIC vs. RMGICR?
- How can the study be sure that the weight loss of samples is due to loss of fluoride ion alone, and not something else, such as soluble monomers in the resins?
- There is no need for color curves in the figures. Change to hatch marks or larger symbols.
- Line 66. Change “discovered” to “developed.”
- Line 77. Delete “of.”
- Sample size discussion. Were there prior measures of the variation of properties in which to calculate the power and sample size?
- Line 114 and others. Mold was made of “silicone rubber,” not “silicon.”
- Line 133. Change “ml” to “mL” throughout. Also, change “370C” to “37°C” throughout.
- Line 144. Define how the fluoride ion at “40ml of 10 PPM solution” was generated.
- Line 149. Should “phage” be “phase”?
- Line 155. What is “NW” in the IBM reference?
- Line 134. What is “TSIAB solution?”
- Line 211. Change “change” to “are changed.”
Author Response
The author appreciates the valuable comments of the reviewer. The comments were helpful in improving the quality of the manuscript.
Comment: Many grammatical and spelling errors, correctable by the authors or the journal’s editorial assistant.
Reply: Thank you for pointing out the typos. The typos and grammatical errors have been revised accordingly.
Comment: A table of materials used should be substituted for the descriptions in the text.
Reply: Thank you for the comment. A table describing the materials has been added to the manuscript.
Comment: Include batch numbers. The restorative material names are confusing: RMGIC vs. RMGICR?
Reply: Thank you for the comment. The batch number has been added to the manuscript. In our study, we have used 2 types of resin-modified glass ionomer (RMGI). The first one was resin-modified glass ionomer liner (RMGIL) and the second one was resin-modified glass ionomer restorative (RMGIR). That’s why there are 3 different abbreviations used in the manuscript. "RMGL" and "RMGR" were used to specify the types, and RMGI was used to address the resin-modified glass ionomer in general. We have revised the abbreviation in the revised manuscript to clear the confusion.
Comment: How can the study be sure that the weight loss of samples is due to loss of fluoride ion alone, and not something else, such as soluble monomers in the resins?
Reply: Thank you for the comment. The author completely agrees with the reviewer. The weight loss of resin-based materials could be multifactorial. The amount of uncured monomers is the major contributor to weight loss and water sorption. We have added these phenomena to the revised manuscript
Comment: There is no need for color curves in the figures. Change to hatch marks or larger symbols.
Reply: Thank you for the comment. The graphs have been revised accordingly
Comment: Line 66. Change “discovered” to “developed.”
Reply: Thank you for pointing out the typos. The errors have been revised accordingly
Comment: Line 77. Delete “of.”
Reply: Thank you for pointing out the typos. The errors have been revised accordingly
Comment: Sample size discussion. Were there prior measures of the variation of properties in which to calculate the power and sample size?
Reply: Thank you for the comment. The sample size calculation section has been revised.
Comment: Line 114 and others. Mold was made of “silicone rubber,” not “silicon.”
Reply: Thank you for pointing out the typos. The errors have been revised accordingly
Comment: Line 133. Change “ml” to “mL” throughout. Also, change “370C” to “37°C” throughout.
Reply: Thank you for pointing out the typos. The errors have been revised accordingly
Comment: Line 144. Define how the fluoride ion at “40ml of 10 PPM solution” was generated.
Reply: Thank you for the comment. The description has been added
Comment: Line 149. Should “phage” be “phase”?
Reply: Thank you for pointing out the typos. The errors have been revised accordingly
Comment: Line 155. What is “NW” in the IBM reference?
Reply: Thank you for pointing out the typos. The errors have been revised accordingly
Comment: Line 134. What is “TSIAB solution?”
Reply: Thank you for the comment. TISAB is a buffer solution used in potentiometric measurements, especially with ion-selective electrodes, to stabilize the ionic strength and pH of a sample. Its main role is to keep the activity coefficient of ions constant, thereby improving the accuracy of measurements like fluoride analysis
Comment: Line 211. Change “change” to “are changed.”
Reply: Thank you for pointing out the typos. The errors have been revised accordingly
Reviewer 3 Report
Comments and Suggestions for Authors
Dear Authors,
Thank you for the opportunity to review your manuscript titled "Fluoride Release and Recharge Properties of Restorative Dental Materials and Its Association with Mass Stability." The study addresses a highly relevant clinical topic in Restorative Dentistry, investigating the fluoride release and recharge properties of restorative materials and their association with mass stability. The methodology employed is well-detailed, and the results provide valuable insights. However, I have identified several points that require clarification or improvement to enhance the quality and clarity of the manuscript. Please address each of the following comments point-by-point in your revision. General CommentsThere is an inconsistency in the abbreviation of materials, particularly for Resin Modified Glass Ionomers (RMGI). In the Abstract, "RMGL" and "RMGR" are used, while in the Results, "RMGIL" and "RMGIR" appear. Please standardize the nomenclature throughout the entire manuscript to avoid confusion.
1. IntroductionThe introduction establishes the importance of fluoride and the research gap regarding the association of fluoride release/recharge with mass stability. It would be beneficial to elaborate a bit more on the clinical relevance of restorative material mass stability in a dental context, right from the introduction. This would help to more robustly justify why mass stability is a crucial component to be investigated alongside fluoride release/recharge.
2. Materials and Methods To ensure the reproducibility of the study, it is fundamental that all materials used are identified with their respective batch numbers. Please add this information for each listed material.The description of the solution for fluoride recharge only mentions "40ml of 10 PPM solution." It is essential to specify which fluoride salt was used to prepare this solution (e.g., sodium fluoride, stannous fluoride, etc.). This information is critical for comparing results with other studies and for validating the method.
Given the focus on mass stability and the potentially hygroscopic nature of some of the tested materials, please describe any environmental controls (e.g., temperature and humidity) maintained during the weight measurements. This would add methodological rigor and transparency to the data collection process.
The statistical analysis section states that "Two-way ANOVA was conducted to evaluate the effect of materials and incubation time on the amount of fluoride release/ recharge and weight loss/ gain." Please clarify whether the interaction effects between factors (material x time) were evaluated, and if so, how they were interpreted and considered in the presentation of the results.
3. ResultsAlthough figures (1 to 6) are referenced to present most of the results, I suggest including complementary tables with the complete mean values and standard deviations for all key measurements (fluoride release at each time point, weight loss, fluoride recharge, weight gain). This would facilitate data consultation and comparison.
In Table 1, the Pearson correlation coefficient is incorrectly labeled as "R²". The R² (coefficient of determination) is always a positive value ranging from 0 to 1 and represents the proportion of explained variance. However, the presented values include negative numbers (e.g., -0.507, -0.827). A negative value indicates that what is being presented is the Pearson correlation coefficient (r), which can range from -1 to 1. Please correct the label in Table 1 from "R²" to "r" so that the notation is statistically correct and reflects the presented values.
4. DiscussionIn the discussion, the authors note that the low fluoride recharge capacity of conventional GIC in their study contrasts with previous literature (reference [35], Naoum et al., 2007). While possible reasons (experimental conditions, material formulations) are mentioned, a more in-depth discussion would be welcome. Could you elaborate on specific differences in protocols (e.g., fluoride concentration in the recharge solution, exposure time) or GIC formulations that might explain this divergence?
The authors highlight the scarcity of previous studies correlating fluoride recharge with weight gain/loss. I suggest emphasizing that, due to this novelty, the conclusions about this specific correlation, while important, open a field for future validations and comparisons with other studies as the literature on the topic expands.
5. Conclusions
The conclusion states that "Fluoride release/ recharge has a mild correlation with the mass stability of restorative dental materials." However, the results show a "strong" correlation between fluoride release and weight loss (R² = 0.724) and a "moderate" correlation (though negative, r = -0.434) for fluoride recharge. I suggest that the conclusion be more specific, distinguishing the different strengths of correlation observed for each pair of variables (release-loss, recharge-gain), rather than a single generalization of "mild correlation."
I believe addressing these points will significantly contribute to the clarity and scientific robustness of your manuscript.
Comments on the Quality of English LanguageThe English language used in the manuscript is generally understandable, conveying the core message of the research. However, there are several areas where improvements could be made to enhance clarity, precision, and overall academic rigor.
Author Response
The author appreciates the valuable comments of the reviewer. The comments were helpful in improving the quality of the manuscript.
Comment: Thank you for the opportunity to review your manuscript titled "Fluoride Release and Recharge Properties of Restorative Dental Materials and Its Association with Mass Stability." The study addresses a highly relevant clinical topic in Restorative Dentistry, investigating the fluoride release and recharge properties of restorative materials and their association with mass stability. The methodology employed is well-detailed, and the results provide valuable insights.
Reply: Thank you for the positive feedback and insight.
Comment: There is an inconsistency in the abbreviation of materials, particularly for Resin Modified Glass Ionomers (RMGI). In the Abstract, "RMGL" and "RMGR" are used, while in the Results, "RMGIL" and "RMGIR" appear. Please standardize the nomenclature throughout the entire manuscript to avoid confusion.
Reply: Thank you for the comment. In our study, we have used 2 types of resin-modified glass ionomer (RMGI). The first one was resin-modified glass ionomer liner (RMGIL) and the second one was resin-modified glass ionomer restorative (RMGIR). That’s why there are 3 different abbreviations used in the manuscript. "RMGL" and "RMGR" were used to specify the types and RMGI was used to address the resin-modified glass ionomer in general. We have revised the abbreviation in the revised manuscript to clear the confusion.
Comment: The introduction establishes the importance of fluoride and the research gap regarding the association of fluoride release/recharge with mass stability. It would be beneficial to elaborate a bit more on the clinical relevance of restorative material mass stability in a dental context, right from the introduction. This would help to more robustly justify why mass stability is a crucial component to be investigated alongside fluoride release/recharge.
Reply: Thank you for the comment. The introduction section has been revised accordingly.
Comment: Materials and Methods To ensure the reproducibility of the study, it is fundamental that all materials used are identified with their respective batch numbers. Please add this information for each listed material.
Reply: Thank you for the comment. The batch number has been added to Table 1.
Comment: The description of the solution for fluoride recharge only mentions "40ml of 10 PPM solution." It is essential to specify which fluoride salt was used to prepare this solution (e.g., sodium fluoride, stannous fluoride, etc.). This information is critical for comparing results with other studies and for validating the method.
Reply: Thank you for the comment. The information has been added to the revised manuscript.
Comment: Given the focus on mass stability and the potentially hygroscopic nature of some of the tested materials, please describe any environmental controls (e.g., temperature and humidity) maintained during the weight measurements. This would add methodological rigor and transparency to the data collection process.
Reply: Thank you for the comment. The information has been added to the revised manuscript.
Comment: The statistical analysis section states that "Two-way ANOVA was conducted to evaluate the effect of materials and incubation time on the amount of fluoride release/ recharge and weight loss/ gain." Please clarify whether the interaction effects between factors (material x time) were evaluated, and if so, how they were interpreted and considered in the presentation of the results.
Reply: Thank you for the comment. The information has been added to the revised manuscript.
Comment: Although figures (1 to 6) are referenced to present most of the results, I suggest including complementary tables with the complete mean values and standard deviations for all key measurements (fluoride release at each time point, weight loss, fluoride recharge, weight gain). This would facilitate data consultation and comparison.
Reply: Thank you for the comment. A table has been added to the revised manuscript with quantitative data.
Comment: In Table 1, the Pearson correlation coefficient is incorrectly labeled as "R²". The R² (coefficient of determination) is always a positive value ranging from 0 to 1 and represents the proportion of explained variance. However, the presented values include negative numbers (e.g., -0.507, -0.827). A negative value indicates that what is being presented is the Pearson correlation coefficient (r), which can range from -1 to 1. Please correct the label in Table 1 from "R²" to "r" so that the notation is statistically correct and reflects the presented values.
Reply: Thank you for pointing out the error. We have corrected the error in the revised manuscript.
Comment: In the discussion, the authors note that the low fluoride recharge capacity of conventional GIC in their study contrasts with previous literature (reference [35], Naoum et al., 2007). While possible reasons (experimental conditions, material formulations) are mentioned, a more in-depth discussion would be welcome. Could you elaborate on specific differences in protocols (e.g., fluoride concentration in the recharge solution, exposure time) or GIC formulations that might explain this divergence?
Reply: Thank you for the comment. The discussion section has been revised accordingly.
Comment: The authors highlight the scarcity of previous studies correlating fluoride recharge with weight gain/loss. I suggest emphasizing that, due to this novelty, the conclusions about this specific correlation, while important, open a field for future validations and comparisons with other studies as the literature on the topic expands.
Reply: Thank you for pointing out an important aspect. We have revised this section as per the suggestion.
Comment: The conclusion states that "Fluoride release/ recharge has a mild correlation with the mass stability of restorative dental materials." However, the results show a "strong" correlation between fluoride release and weight loss (R² = 0.724) and a "moderate" correlation (though negative, r = -0.434) for fluoride recharge. I suggest that the conclusion be more specific, distinguishing the different strengths of correlation observed for each pair of variables (release-loss, recharge-gain), rather than a single generalization of "mild correlation."
Reply: Thank you for pointing out the error. We have revised this section accordingly.
Comment: The English language used in the manuscript is generally understandable, conveying the core message of the research. However, there are several areas where improvements could be made to enhance clarity, precision, and overall academic rigor.
Reply: Thank you for the comment. The language and grammar have been carefully revised.
Reviewer 4 Report
Comments and Suggestions for Authors
General Assessment:
This manuscript presents an in-vitro study evaluating the fluoride release and recharge capacity of different restorative dental materials and examines how these properties correlate with the materials’ mass stability. The topic is timely and relevant, considering the ongoing development of fluoride-releasing materials for caries prevention. The experimental approach is generally well-conceived and the statistical methods are appropriate for the study objectives.
However, the manuscript requires significant revision before being suitable for publication. Issues related to grammar, clarity, scientific accuracy, and consistency in terminology must be addressed to improve its readability and professional quality.
Major Comments:
Title and Abstract:
The title should be revised to improve clarity and grammar. Suggested:
“Fluoride Release, Recharge, and Mass Stability of Restorative Dental Materials: An In Vitro Study”
The abstract has multiple grammatical issues and redundancies. Revise for clarity and conciseness. For instance:
“The objective of this study was to evaluate the fluoride release and recharge capacity of restorative dental materials and their correlation with mass stability.”
Introduction:
The historical context of dental materials is informative but should be more concise to keep focus on the study’s relevance.
Several sentences are repetitive or grammatically awkward. For example:
“The desire for permanent restorative material that satisfies both esthetic and functional properties keep the researcher and manufacturer quest for new restorative material” →
Should be: “The ongoing demand for restorative materials that combine esthetic appeal and functional longevity continues to drive research and development.”
The rationale for correlating fluoride behavior with mass stability should be better justified with references.
Materials and Methods:
The description of material groups (e.g., GIC, RMGI-R, RMGI-L, etc.) needs better standardization and definition. Avoid inconsistencies like RMGL/RMGIL/RMGICL.
The methodology for fluoride measurement and weight assessment is valid, but some steps (e.g., calibration, sample preparation) lack detail.
Units and concentrations should be standardized throughout the section (e.g., “10 PPM” → “10 ppm”).
Results:
Figures are mentioned but not included in the peer review document. Ensure these are legible and well-captioned.
The presentation of statistical data is sound, but descriptions of trends should be better connected to clinical relevance.
Clarify inconsistencies in group naming (RMGIR vs. RMGICR) and check for typographical errors (e.g., “weas” instead of “was”).
Discussion:
This section presents meaningful interpretation, especially in linking fluoride behavior with structural degradation.
It would benefit from more critical analysis comparing findings to previous studies. The contrast with Naoum et al. (2007), for example, deserves further elaboration.
Consider discussing potential implications for pediatric versus adult patients, or high-risk versus low-risk caries populations.
Conclusion:
Well summarized but should emphasize clinical translation and limitations more clearly.
Minor Comments:
Grammar and syntax issues are frequent throughout the manuscript. Consider professional language editing.
The term “recharge phage” should be corrected to “recharge phase.”
Avoid inconsistent or undefined abbreviations (e.g., “RMGL” vs “RMGIL”).
Recommendation:
Major Revision
Suggestions for Improvement (Summary):
Language editing for grammar, clarity, and technical precision.
Standardize terminology and abbreviations for all material groups.
Enhance the abstract and introduction for scientific focus.
Improve figure labeling and ensure visual clarity.
Deepen the discussion with critical engagement of contrasting literature.
Clarify implications for clinical practice and future research directions.
Author Response
The author appreciates the valuable comments of the reviewer. The comments were helpful in improving the quality of the manuscript.
Comment: This manuscript presents an in-vitro study evaluating the fluoride release and recharge capacity of different restorative dental materials and examines how these properties correlate with the materials’ mass stability. The topic is timely and relevant, considering the ongoing development of fluoride-releasing materials for caries prevention. The experimental approach is generally well-conceived and the statistical methods are appropriate for the study objectives.
Reply: Thank you for the positive feedback and insight.
Comment: The title should be revised to improve clarity and grammar. Suggested: “Fluoride Release, Recharge, and Mass Stability of Restorative Dental Materials: An In Vitro Study”
Reply: Thank you for the comment. The title has been revised
Comment: The abstract has multiple grammatical issues and redundancies. Revise for clarity and conciseness. For instance: “The objective of this study was to evaluate the fluoride release and recharge capacity of restorative dental materials and their correlation with mass stability.”
Reply: Thank you for the comment. The abstract has been revised to correct the grammatical errors
Comment: The historical context of dental materials is informative but should be more concise to keep focus on the study’s relevance.
Reply: Thank you for the comment. This section has been revised to make it more concise.
Comment: Several sentences are repetitive or grammatically awkward. For example: “The desire for permanent restorative material that satisfies both esthetic and functional properties keep the researcher and manufacturer quest for new restorative material” →
Should be: “The ongoing demand for restorative materials that combine esthetic appeal and functional longevity continues to drive research and development.”
Reply: Thank you for the comment. The sentence has been revised accordingly.
Comment: The rationale for correlating fluoride behavior with mass stability should be better justified with references.
Reply: Thank you for the comment. The rationale has been addressed.
Comment: The description of material groups (e.g., GIC, RMGI-R, RMGI-L, etc.) needs better standardization and definition. Avoid inconsistencies like RMGL/RMGIL/RMGICL.
Reply: Thank you for the comment. The abbreviation has been revised.
Comment: The methodology for fluoride measurement and weight assessment is valid, but some steps (e.g., calibration, sample preparation) lack detail.
Reply: Thank you for the comment. The detail has been added to the revised manuscript.
Comment: Units and concentrations should be standardized throughout the section (e.g., “10 PPM” → “10 ppm”).
Reply: Thank you for the comment. The unit has been corrected in the revised manuscript.
Comment: Figures are mentioned but not included in the peer review document. Ensure these are legible and well-captioned.
Reply: Thank you for the comment. The figures are presented within the manuscript file. As we have used the journal provided template. Separate files for each figure are not required by the journal.
Comment: The presentation of statistical data is sound, but descriptions of trends should be better connected to clinical relevance.
Reply: Thank you for the comment. The clinical relevance of the obtained results has been added to the discussion section
Comment: Clarify inconsistencies in group naming (RMGIR vs. RMGICR) and check for typographical errors (e.g., “weas” instead of “was”).
Reply: Thank you for pointing out the typos. The errors have been revised accordingly
Comment: The discussion section presents meaningful interpretation, especially in linking fluoride behavior with structural degradation. It would benefit from more critical analysis comparing findings to previous studies. The contrast with Naoum et al. (2007), for example, deserves further elaboration.
Reply: Thank you for the comment. This section has been revised accordingly
Comment: Consider discussing potential implications for pediatric versus adult patients, or high-risk versus low-risk caries populations.
Reply: Thank you for the suggestion. This section has been revised accordingly
Comment: The conclusion is well summarized but should emphasize clinical translation and limitations more clearly.
Reply: Thank you for the suggestion. This section has been revised accordingly
Comment: Grammar and syntax issues are frequent throughout the manuscript. Consider professional language editing.
Reply: Thank you for the comment. The grammar and syntax have been revised carefully.
Comment: The term “recharge phage” should be corrected to “recharge phase.”
Reply: Thank you for pointing out the typos. The errors have been revised accordingly
Comment: Avoid inconsistent or undefined abbreviations (e.g., “RMGL” vs “RMGIL”).
Reply: Thank you for the comment. The abbreviations have been revised accordingly
Round 2
Reviewer 1 Report
Comments and Suggestions for Authors
The authors have addressed the previous comments thoroughly, and the manuscript has improved substantially.
However, there is one last point that requires clarification. In the initial version of the manuscript, the effect size for the sample size calculation was reported as f = 0.75, whereas in the revised version it now appears as f = 0.60. Could the authors please clarify why this value was changed?
Author Response
Thank you for the positive and valuable feedback. The comments were helpful in improving the quality of the manuscript.
Comment: The authors have addressed the previous comments thoroughly, and the manuscript has improved substantially.
Reply: Thank you for the positive feedback.
Comment: However, there is one last point that requires clarification. In the initial version of the manuscript, the effect size for the sample size calculation was reported as f = 0.75, whereas in the revised version, it now appears as f = 0.60. Could the authors please clarify why this value was changed?
Reply: Thank you for the comment. It was a typo in the initial version. We have rechecked and confirmed the effect size. The sample size n=10 was calculated using the effect size f=0.060. Sorry for the unintentional error.
Reviewer 2 Report
Comments and Suggestions for Authors
- The title change and addition of Table 1 are welcome additions.
- The author’s comments were very complementary but some of the corrections were not made.
- Since a power analysis is presented, it still does not present preliminary data from an earlier study from which to estimate variation in measurements, which then is used to determine minimum sample size in the current experiment. This is a disqualifying flaw.
- The samples were allowed to soak in sessile water rather than running water. Restorative materials experience a constant flow of saliva, liquids and solids in the mouth. A more appropriate experimental plan would have been to store the restorative materials in flowing water and then measure fluoride loss or gain (from recharging) at the end of the four time periods.
- Line 28, make it “resin-based composite” because there are metal- and ceramic-based composites. “Composite” is an adjective.
- No need for color in Figures 1, 3, and 4, unless publisher disagrees.
- Line 59, “ restorations.
- Line 69, “…and wear resistance…
- Line 72 ”…patents…”
- Line 75, why was the reference to Wilson & Kent dropped in Ref. 26?
- Line 80, “…properties,…cements…”
- Line 138, “…silicone…”
- Line 166, “…replace “was” with “”
- Line 172, “ using the electrode…”
- Line 184, “…comparisons…”
- Line 186, “…weight…”
- Line 192, “…until instead of till…”
- Lines 216, 224, 233, 243, & 257, add “ letter” after “alphabet…”
- Line 321, “…that is exchanged…”
- Line 350, “…sites while…”
Author Response
Thank you for the positive and valuable feedback. The comments were helpful in improving the quality of the manuscript.
Comment: The title change, and addition of Table 1 are welcome additions. The author’s comments were very complementary but some of the corrections were not made.
Reply: Thank you for the positive feedback.
Comment: Since a power analysis is presented, it still does not present preliminary data from an earlier study from which to estimate variation in measurements, which then is used to determine minimum sample size in the current experiment. This is a disqualifying flaw.
Reply: Thank you for the comment. As we explained in the revised version, a pilot study was carried out to estimate the effect size. We agree with the reviewer that the effect size should be calculated from a previously published article. However, as we explain in the discussion section, the quantifying method of fluoride release and recharge has not been standardized, we decided to conduct a pilot study using the same equipment and the same setup. We believe that the effect size obtained from previously published articles may vary, as different electrodes and setups were used in those studies.
Comment: The samples were allowed to soak in sessile water rather than running water. Restorative materials experience a constant flow of saliva, liquids and solids in the mouth. A more appropriate experimental plan would have been to store the restorative materials in flowing water and then measure fluoride loss or gain (from recharging) at the end of the four time periods.
Reply: Thank you for explaining the logical approach that should have been taken to mimic the clinical condition. We completely agree with the reviewer. However, thinking practically, it is not possible to mimic exact clinical conditions. The quantification was made by measuring the storage media. In the case of running water, it would be difficult to detect and quantify. We have highlighted this limitation in the discussion section.
Comment: Line 28, make it “resin-based composite” because there are metal- and ceramic-based composites. “Composite” is an adjective.
Reply: Thank you for pointing out the typo. It has been revised accordingly
Comment: No need for color in Figures 1, 3, and 4, unless publisher disagrees.
Reply: Thank you for the comment. Each color in figures 1,3, and 4 represents/identifies the tested materials. It would be difficult to identify the groups in the figures in RGB mode. However, it can be changed to RGB mode at any stage of processing.
Comment: Line 59, “ restorations.
Reply: Thank you for pointing out the typo. It has been revised accordingly
Comment: Line 69, “…and wear resistance…
Reply: Thank you for pointing out the typo. It has been revised accordingly
Comment: Line 72 ”…patents…”
Reply: Thank you for pointing out the typo. It has been revised accordingly
Comment: Line 75, why was the reference to Wilson & Kent dropped in Ref. 26?
Reply: Thank you for the comment. The citation for this statement remains unchanged (ref 13, 14). A new reference has been added (ref-26) to replace an outdated one as per one of the reviewers’ comments.
Comment: Line 80, “…properties,…cements…”
Reply: Thank you for pointing out the typo. It has been revised accordingly
Comment: Line 138, “…silicone…”
Reply: Thank you for pointing out the typo. It has been revised accordingly
Comment: Line 166, “…replace “was” with “”
Reply: Thank you for pointing out the typo. It has been revised accordingly
Comment: Line 172, “ using the electrode…”
Reply: Thank you for pointing out the typo. It has been revised accordingly
Comment: Line 184, “…comparisons…”
Reply: Thank you for pointing out the typo. It has been revised accordingly
Comment: Line 186, “…weight…”
Reply: Thank you for pointing out the typo. It has been revised accordingly
Comment: Line 192, “…until instead of till…”
Reply: Thank you for pointing out the typo. It has been revised accordingly
Comment: Lines 216, 224, 233, 243, & 257, add “ letter” after“alphabet…”
Reply: Thank you for pointing out the typo. It has been revised accordingly
Comment: Line 321, “…that is exchanged…”
Reply: Thank you for pointing out the typo. It has been revised accordingly
Comment: Line 350, “…sites while…”
Reply: Thank you for pointing out the typo. It has been revised accordingly
Reviewer 3 Report
Comments and Suggestions for Authors
The manuscript addresses a clinically highly relevant topic, investigating fluoride release, recharge capability, and mass stability of various restorative dental materials. The employed methodology, with systematic in vitro measurements and comprehensive statistical analysis, is pertinent to the research question. The objective is clear, and the results hold significant potential value for guiding material selection in dental practice, especially concerning longevity and caries prevention.
The article has progressed significantly with the revisions the team carried out. The improvements in clarity, grammatical correction, and text fluidity are remarkable and demonstrate your very dedicated work. The manuscript is visibly more robust and professional. However, in its current form, it still requires further revisions. I notice that a few specific phrases and terms could still be refined. It is possible that the excessive use of "track changes" has hindered the visualization of every meticulous adjustment, resulting in the persistence of minor inconsistencies or typos. Some constructions, although improved, could still be more concise and direct to optimize the academic flow and impact. My suggestion for this final phase is that you incorporate all changes and create a "clean" version of the article, without visible "track changes." I believe that, from that point on, a new review focused solely on small optimizations, like those I mentioned earlier, will be much more efficient. In addiction, the lack of references in certain sections of the discussion and in more general statements within the introduction is a critical point that needs to be addressed. It is fundamental that any statement that is not a direct and original result of your study (presented in the Results section), or that is not universally accepted knowledge (which is rare in scientific articles), must be properly referenced. This includes, in the Introduction, that when you mention "several studies" or "past research has revealed," it is imperative that references for these studies are included immediately after the statement. This ensures that the foundational information for your work is supported by evidence and that the reader can verify the source. Similarly, in the Discussion, although it is the place to interpret your own results, it is also where you compare them with existing literature, contextualize them, and explain why certain phenomena occur. Whenever you make a claim about what is known in the field, about other researchers' findings, or about the mechanism of something, a reference must accompany it. This is necessary because including references validates your assertions, showing that they are based on prior research and are not merely opinions, thereby significantly elevating the level of evidence and the robustness of your argument. It also provides transparency, allowing readers and reviewers to trace the origin of information, delve deeper into the topic if they wish, and evaluate the validity of the data you are using as a basis. Furthermore, it is the correct way to avoid plagiarism by giving credit to the work of other researchers, and it helps position your study within the existing body of knowledge, demonstrating how it connects to and contributes to the field. My suggestion is to please conduct a thorough review of the entire Introduction and Discussion sections. For every sentence or paragraph that presents information not directly from your own study, or that generalizes about knowledge in the field, add the relevant reference(s). If you mention "several studies," try to cite the most representative studies or the literature reviews that compile this information. This addition of references will make your manuscript much more solid, scientifically rigorous, and reliable for publication. Comments on the Quality of English LanguageBased on the observed revisions and text analysis, it is clear that the article has undergone an intensive editing process, characterized by an excessive amount of "track changes." While these changes aimed to improve the content, the sheer volume of markups might have contributed to the persistence of certain linguistic issues. Specifically, the English in the manuscript still features phrases that need improvement, sounding awkward or unnatural for academic English, which can hinder reading fluidity. Despite multiple revisions, typos remain scattered throughout the text. Furthermore, in certain passages, the absence of a main verb or the construction of incomplete sentences compromises grammar and clarity of the message. These points suggest that a final review, ideally on a "clean" version of the text, would be crucial to eradicate these minor imperfections.
Author Response
Thank you for the positive and valuable feedback. The comments were helpful in improving the quality of the manuscript.
Comment: The manuscript addresses a clinically highly relevant topic, investigating fluoride release, recharge capability, and mass stability of various restorative dental materials. The employed methodology, with systematic in vitro measurements and comprehensive statistical analysis, is pertinent to the research question. The objective is clear, and the results hold significant potential value for guiding material selection in dental practice, especially concerning longevity and caries prevention. The article has progressed significantly with the revisions the team carried out. The improvements in clarity, grammatical correction, and text fluidity are remarkable and demonstrate your very dedicated work.
Reply: Thank you for the positive feedback.
Comment: The manuscript is visibly more robust and professional. However, in its current form, it still requires further revisions. I notice that a few specific phrases and terms could still be refined. It is possible that the excessive use of "track changes “has hindered the visualization of every meticulous adjustment, resulting in the persistence of minor inconsistencies or typos. Some constructions, although improved, could still be more concise and direct to optimize the academic flow and impact. Suggestion for this final phase is that you incorporate all changes and create a "clean" version of the article, without visible "track changes." I believe that, from that point on, a new review focused solely on small optimizations, like those I mentioned earlier, will be much more efficient.
Reply: Thank you for the valuable suggestion. We have uploaded the revised version of the manuscript, both with and without track changes
Comment: In addition, the lack of references in certain sections of the discussion and in more general statements within the introduction is a critical point that needs to be addressed. It is fundamental that any statement that is not a direct and original result of your study (presented in the Results section), or that is not universally accepted knowledge (which is rare in scientific articles), must be properly referenced. This includes, in the Introduction, that when you mention "several studies" or "past research has revealed," it is imperative that references for these studies are included immediately after the statement. This ensures that the foundational information for your work is supported by evidence and that the reader can verify the source. Similarly, in the Discussion, although it is the place to interpret your own results, it is also where you compare them with existing literature, contextualize them, and explain why certain phenomena occur. Whenever you make a claim about what is known in the field, about other researchers' findings, or about the mechanism of something, a reference must accompany it. This is necessary because including references validates your assertions, showing that they are based on prior research and are not merely opinions, thereby significantly elevating the level of evidence and the robustness of your argument. It also provides transparency, allowing readers and reviewers to trace the origin of information, delve deeper into the topic if they wish, and evaluate the validity of the data you are using as a basis. Furthermore, it is the correct way to avoid plagiarism by giving credit to the work of other researchers, and it helps position your study within the existing body of knowledge, demonstrating how it connects to and contributes to the field. My suggestion is to please conduct a thorough review of the entire Introduction and Discussion sections. For every sentence or paragraph that presents information not directly from your own study, or that generalizes about knowledge in the field, add the relevant reference(s). If you mention "several studies," try to cite the most representative studies or the literature reviews that compile this information. This addition of references will make your manuscript much more solid, scientifically rigorous, and reliable for publication.
Reply: Thank you for the valuable suggestion. We have revised the citation accordingly
Comment: Based on the observed revisions and text analysis, it is clear that the article has undergone an intensive editing process, characterized by an excessive amount of "track changes." While these changes aimed to improve the content, the sheer volume of markups might have contributed to the persistence of certain linguistic issues. Specifically, the English in the manuscript still features phrases that need improvement, sounding awkward unnatural for academic English, which can hinder reading fluidity. Despite multiple revisions, typos remain scattered throughout the text. Furthermore, in certain passages, the absence of a main verb or the construction of incomplete sentences compromises grammar and clarity of the message. These points suggest that a final review, ideally on a "clean" version of the text, would be crucial to eradicate these minor imperfections.
Reply: Thank you for the valuable suggestion. We have made another round of grammar and typo check without the track changes version to eliminate those errors.